# Pathomorphological Diagnostic Criteria for Focal Cortical Dysplasias and Other Common Epileptogenic Lesions—Review of the Literature

**DOI:** 10.3390/diagnostics13071311

**Published:** 2023-03-31

**Authors:** Dimitar Metodiev, Krassimir Minkin, Margarita Ruseva, Rumiana Ganeva, Dimitar Parvanov, Sevdalin Nachev

**Affiliations:** 1Neuropathological Laboratory, University Hospital “Saint Ivan Rilski”, 1431 Sofia, Bulgaria; 2Department of Clinical Pathology, Nadezhda Women’s Health Hospital, 1373 Sofia, Bulgaria; 3Epilepsy Surgery Center, University Hospital “Saint Ivan Rilski”, 1431 Sofia, Bulgaria; 4Department of Research, Nadezhda Women’s Health Hospital, 1373 Sofia, Bulgaria

**Keywords:** focal cortical dysplasia, pharmacoresistant epilepsy, mild malformation of cortical development

## Abstract

Focal cortical dysplasia (FCD) represents a heterogeneous group of morphological changes in the brain tissue that can predispose the development of pharmacoresistant epilepsy (recurring, unprovoked seizures which cannot be managed with medications). This group of neurological disorders affects not only the cerebral cortex but also the subjacent white matter. This work reviews the literature describing the morphological substrate of pharmacoresistant epilepsy. All illustrations presented in this study are obtained from brain biopsies from refractory epilepsy patients investigated by the authors. Regarding classification, there are three main FCD types, all of which involve cortical dyslamination. The 2022 revision of the International League Against Epilepsy (ILAE) FCD classification includes new histologically defined pathological entities: mild malformation of cortical development (mMCD), mild malformation of cortical development with oligodendroglial hyperplasia in frontal lobe epilepsy (MOGHE), and “no FCD on histopathology”. Although the pathomorphological characteristics of the various forms of focal cortical dysplasias are well known, their aetiologic and pathogenetic features remain elusive. The identification of genetic variants in FCD opens an avenue for novel treatment strategies, which are of particular utility in cases where total resection of the epileptogenic area is impossible.

## 1. Introduction

The morbidity of active epilepsy is similar for countries with different economic development; it is 2.3–5.9/1000 for high-income countries, 3.7–13.3/1000 for high-income to middle-income economies, 2.4–2.8/1000 for countries with low-income to middle-income economies, and 3.6–15.4/1000 for low-income economies [1]. The epilepsy begins before the age of 16 in 50–60% of patients [1]. Despite the increasing number of antiepileptic drugs (AED) available, the percentage of non-responders remains consistent—between 20 and 25% of all epilepsy cases. Only about one-quarter of these pharmacoresistant patients are suitable candidates for neurosurgery [1].

According to the latter definition proposed by ILAE, 2009, “drug resistant patients are those who have failed an adequately administered, appropriately selected and well tolerated 2 AED, in monotherapy or in combination, to achieve prolonged freedom from seizures (for a period of 1 year to 3 times longer than the longest interictal period prior to the last year)” [2].

Numerous types of surgical interventions aimed at managing epileptic episodes are available. They are grouped in three main categories: epileptogenic focus resection (lesionectomy, zonectomy, amygdalohippocampectomy, lobectomy, radiosurgical destruction, thermocoagulation), disconnection of the epileptogenic region from the rest of the brain (disconnection of hypothalamic hamartoma, multiple subpial transsection, corpus callosotomy), and epilepsy zone desynchronisation through chronic electrical stimulation (deep brain stimulation, hippocampal stimulation, left vagus nerve stimulation). The most common surgical type is focal resection. The effectiveness of the operation can be evaluated in terms of several factors: reduction of epileptic seizures frequency and severity, quality of life improvement, alterations in the doses or number of AEDs necessary, post-surgery complications, etc. The optimum outcome would be complete seizure freedom. Surgery is also the only definitive way to diagnose the specific type of epilepsy lesion—through subsequent histological assessment of the resected brain specimen [3].

According to the revised ILAE classification scheme for FCD, there are diverse categories of malformative anomalies which comprise the majority of histopathological findings and tissue features [4]. The first category consists of vertical dyslamination, including microcolumns and heterotopic neurones (FCD Ia), horizontal dyslamination (FCD Ib), and a combination of the two types of dyslamination of the neocortex (FCD Ic). The second group includes cases with cortical dyslamination and dysmorphic neurones without (FCD IIa) or with balloon cells (FCD IIb). The third category is defined by cortical dyslamination and other classic epileptogenic lesions—hippocampal sclerosis (FCD IIIa), tumours (FCD IIIb), cerebrovascular malformations (FCD IIIc), or other early acquired epileptogenic lesions (traumatic brain injury, inflammatory processes, FCD IIId). The 2022 revised ILAE classification of FCD includes new pathological entities predominantly affecting the white matter. The detection of subcortical heterotopic neurones in the absence of other structural lesion is defined as mild malformation of cortical development (mMCD) [4]. The presence of heterotopic neurones, in combination with oligodendroglial hyperplasia and blurring of the grey–white matter boundary are defining features of the recently defined condition MOGHE [5]. A separate category encompassing all cases with indefinable epileptogenic lesion has also been introduced and is called “no FCD on histopathology” [4]. The ILAE revision enabled the classification of cases (with abnormal MRI findings) that could not be assigned to any form of FCD according to the previous 2011 classification. These cases used to be defined as FCD not otherwise specified (FCD NOS). Precise diagnosis integrating various types of examination (imaging, histology, and genetic testing) is of clinical relevance as it guides subsequent management of FCD patients.

## 2. Brief History of Focal Cortical Dysplasia

FCD is a diverse group of disorders which could predispose the development of pharmacoresistant epilepsy [6]. The term “dysplasia” typically refers to precancerous conditions (e.g., intestinal or gastric epithelial dysplasia, etc.). Sometimes it is used to describe disorders of developmental character, or abnormal mature cells in terms of size, form, or structural organisation. FCD includes various disturbances of the normal brain development such as cortical dyslamination, disturbed histoarchitectronics, and abnormalities of the subjacent white matter (e.g., heterotopic neurones) [7,8].

The brief history of FCD marked its origin back in 1971 when Taylor et al. reported a case series of 10 patients with pharmacoresistant epilepsy, in whose resected brain specimens they found cortical dyslamination, large neurones with bizarre morphology, and ballooned cells [9]. All of these features are part of the current classification of FCD. In the original work, the authors defined these characteristics as, “Focal dysplasia of the cerebral cortex”. It is noteworthy that even before this report, Crome (1957) identified “dysmorphic neurones” and “giant nerve cells” in the lesion sections from three reported refractory epilepsy patients [10].

Since then, numerous authors have used varied terminology to describe what is now known as FCD. Although the condition’s aetiology remains obscure, it has now been widely accepted that the majority of lesions are associated with disruptions of cortical development [11]. In 1992, Meecke and Veith introduce the term “microdysgenesis” to define focal malformations in the brain parenchyma of generalised epilepsy patients. In the studied subjects, they described characteristic morphological alterations, which largely overlap with the current FCD classification. In 1997, Barkovic et al. used the term “malformation of cortical development” to describe cases of children with mental retardation and young epilepsy patients [11].

In 2004, Palmini et al. put forward recommendations for FCD classification [8], namely: (1) specific terminology related to the different types of changed cells found in the brain parenchyma of FCD patients; (2) new descriptive terms defining the lesions formerly comprising “microdysgenesis”, which as a term itself is no longer used; (3) in-depth classification of histopathological characteristics included in the broad “FCD” term. They defined lesions as either mild malformation of cortical development (mMCD) or FCD. In addition, each category was also divided into two subcategories—type I and II [8].

The latest classification system by Najm et al. (2022) includes not only cortical lesions but also ones localised in the white matter (mMCD and MOGHE) [4].

## 3. Isolated Forms of FCD with Vertical, Horizontal, or a Combination of the Two Types of Cortical Dyslamination (FCD Type I)

FCD type I is an isolated form characterised by histoarchitectural disruption of the six-layer structure of the cerebral cortex (otherwise known as dyslamination). It affects one or more lobes and is predominantly found in young patients with severe epilepsy and psychomotor retardation [12]. There are three subtypes according to the type of histological changes: FCD type Ia defined by vertical dyslamination with microcolumnar organisation and heterotopic neurones; FCD type Ib characterised by horizontal dyslamination; and FCD type Ic with vertical and horizontal dyslamination [4].

The defining features of FCD type Ia resemble structures characteristic of embryonic cortical development [12,13]. By definition, microcolumns are clusters of more than eight vertically oriented neurones (Figure 1A) [12,14]. It predominantly affects the posterior quadrant of the brain [4].

In addition to cortical dyslamination, other structural hallmarks of all FCD I subtypes (Ia, Ib, and Ic) are an increased number of heterotopic neurones found within the white matter, especially in cases of FCD type Ia, as well as hypertrophic neurones outwith layer V of the neocortex [4]. Immunohistochemistry staining for neuronuclear antigen (NeuN) allows visualisation of the abnormal histoarchitectronics (Figure 1B). The same marker can be used for IHC verification of heterotopic neurones in the adjacent white matter [4].

According to the literature, FCD Ia is the most common FCD I subtype. It can be regarded as maturational arrest of cortical development processes mid-gestation, before the histological transition from radial to tangential cortical migration is complete [14].

The aetiology for this type of cortical dyslamination remains unclear. The majority of FCD I cases are sporadic in nature; however, familial forms of the condition have been described suggesting possible genetic aberrations [15]. Maturational arrest features have been described in cases with genetic mutations and chromosomopathies, e.g., in DiGeorge syndrome (22q11.1 locus) [14].

There are various forms of horizontal cortical dyslamination (FCD Ib). One such form presents with reduced number of neurones in the external pyramidal layer (cortical layer III) and calretinin-positive residual ganglion cells. In other cases NeuN-negative cells are detected in the superficial layers. The same neurones do not exhibit pathological features when stained with Hemalaun–Eosin (HE) [7].

In many FCD type Ib lesions there is a relatively increased number of GABA-ergic interneurones positive for calretinin, which reach the neocortical plate via tangential migration. This is likely caused by selective loss of neurones which should have arrived to their final position via radial migration. Identical features to FCD Ib—conserved number of calretinin-positive neurones and reduced number of calretinin-negative pyramidal neurones—have also been described in Ammon’s horn sclerosis in paediatric epilepsy patients [16,17]. Nonetheless, the exact physiological mechanism of deficient GABA-ergic interneurone inhibition contributing to chronic epilepsy remains poorly understood [7].

FCD type Ic is a rare form of FCD combining the distinctive aberrant radial and tangential lamination patterns seen in Ia and Ib, respectively (Figure 1C,D). The pathological features resembling those of FCD Ic in combination with vascular malformations or other acquired developmental conditions (perinatal infarcts) are classified as FCD type IIIc or IIId, respectively [11].

## 4. Isolated Forms of FCD with Cortical Dyslamination and Dysmorphic Neurones with or without Balloon Cells (FCD Type II)

FCD type II consists of two subtypes: FCD IIa—characterised by cortical dyslamination with dysmorphic neurones and absence of balloon cells; FCD IIb—defined by the same features as IIa but with the presence of balloon cells [7].

Indistinguishable cortical layers except for lamina molecularis are distinctive cortical dyslamination features of FCD II. In addition, typical for the condition are also a blurred grey–white matter border and increased density of heterotropic neurones in the white matter. Dysmorphic neurones can be detected using routine HE (Figure 2A) or Cresyl violet (Nissl) staining. Due to accumulation of neurofilament proteins in the bodies of these neurones, immunostaining with antibodies against them (e.g., 2F11 or SMI32) could be of use for verification purposes (Figure 2C) [7].

Balloon cells, which are also seen in the tubers of patients with tuberous sclerosis, can be identified using routine HE staining (Figure 2B). To definitively establish their presence, additional staining with vimentin (Figure 2D) and GFAP-δ is necessary [18]. Some balloon cells also express CD34 and the glial adhesion molecule AMOG [19].

FCD type II is distinguishable from types I and III due to the presence of dysmorphic cells. It is noteworthy that both dysmorphic cells and cells with completely normal morphology are detected. The abnormal cells exhibit characteristic megalocytic (untypically large size) and dysmorphic (abnormal form of body and processes, peripherally located atypical nuclei, abnormal distribution of the Nissl substance) features. On the other hand, balloon cells have large, often eccentric vesicular nuclei and abundant eosinophilic cytoplasm. Binuclear cells are sometimes also found. These abnormal cells may align in vertically oriented microcolumns which are typical for FCD type Ia [7].

The aetiology of FCD II may involve post-zygotic somatic mutations affecting the undifferentiated neuroepithelial cells. Therefore, in this pathological condition, the cortex contains intermixed cell populations composed of normal neuroblasts and abnormal ones arisen due to genetic aberrations [20].

The cellular dysmorphism and abnormal size of FCD IIa and IIb neurones indicate a perturbation at a very early stage of cell differentiation, likely involving cytoskeletal proteins. One such protein associated with microtubules is tau, which is known to be overactivated (in a hyperphosphorylated state) not only in neurodegenerative conditions [21], but also in infant tauopathies—tuberous sclerosis, hemimegaloencephaly, FCD type II, etc. All these early childhood conditions involve disruptions in the mTOR signalling pathway [22]. Multiple key molecules involved in that pathway carrying somatic mutations have been identified in specimens from FCD II patients [23]. One such mutation that seems to have a pivotal role for FCD II is a pathogenic variant in the DEPDC5 gene, which causes disinhibition of the mTOR pathway, leading to increased cellular growth and proliferation [24]. Some selective pharmacological mTOR antagonists seem to offer a promising therapeutic avenue [25]. A recent small clinical trial testing the safety and efficacy of one such mTOR inhibitor (everolimus) in terms of reducing seizure frequency in FCD II patients has completed phase II, and the results are expected to be published soon (ClinicalTrials.gov ID: NCT03198949). Further clinical trials of FDA-approved mTOR inhibitors such as temsirolimus, rapamycin, and sirolimus warrant future investigation.

## 5. Hippocampal Sclerosis and FCD Associated with Hippocampal Sclerosis (FCD IIIa)

Hippocampal sclerosis (HS) is the most common pathomorphological finding in adult patients with drug-resistant temporal lobe epilepsy [4,26,27].

In a European cohort study of 9523 patients with epilepsy who had undergone surgical intervention, HS was diagnosed in 36.4%. In 5% of those cases, an additional pathology was identified—cortical malformations (FCD IIa or Iib), epileptogenic tumours, cerebrovascular lesions, or inflammatory features [27].

The histological fingerprint of HS includes prominent loss of pyramidal cells in the CA1 (Sommer’s sector), CA3, and CA4 hippocampal regions, while the CA2 pyramidal cells and the granular cells of the dentate gyrus remain relatively unaffected [27,28]. The neuropeptide Y- and somatin-immunoreactive interneurones and hilar mossy cells may also be affected [29]. Neuronal loss is associated with reactive astrogliosis (Figure 3) which leads to tissue densification also known as Ammon’s horn sclerosis [30]. The mechanism for selective neuronal loss affecting otherwise identical neural cell populations remains a topic of particular interest for researchers in the field. Leading working theories implicate emerging abnormal neuronal circuits, molecular rearrangement of an ion channel, and neurotransmitter receptor expression as candidates for explaining this phenomenon [31].

Morphological changes in the dentate gyrus, such as dispersed granular cells, are found in most examined hippocampal resections [32]. Neuronal loss is sometimes also seen in adjacent cortical regions, such as the subiculum, the entorhinal cortex, and the amygdala. Moreover, cortical dyslamination features and an increased number of ectopic neurones in the white matter of the temporal lobe are detected [33]. An increased number of “string” blood vessels can also be found whose endothelial cells cannot be visualised, which indicates damage to the capillaries. They are only composed of their basement membranes, experimentally proven by immunohistochemical examination with collagen type IV [34].

ILAE has created a semiquantitative classification method for determining three histologically distinct HS variants [35]. HS type 1, accounting for 70% of the cases, is characterised by prominent neuronal loss and fibrillary astrogliosis in CA4 and CA1 regions, as well as moderate neuronal loss and astrogliosis in CA3 and CA2, and histoarchitectural remodelling of the granular cell layer [35]. HS types 2 and 3 have a rare occurrence. Type 2 HS is defined by marked neuronal loss in CA1 (Sommer’s sector), and in HS type 3 the granular cell layer and CA4 are predominantly affected [35]. This classification system also has clinical implications—patients with HS types 1 and 3 of the dominant hemisphere exhibit verbal memory deficits [36,37]. Type 3 often manifests with dual pathology (HS associated with a second extrahippocampal epileptogenic lesion), whereas type 1 is typically found in patients with history of febrile seizures during early childhood [38].

## 6. Long-Term Epilepsy-Associated Tumours (LEAT). FCD Associated with a Tumour Process (FCD Type IIIb)

According to some authors, the incidence of tumours in patients diagnosed with epilepsy reaches up to 30% [39,40]. It is well known that every slowly growing tumour (e.g., meningioma or glioma) could lead to focal epilepsy; however, the majority of tumours inducing epileptic seizures are glioneuronal [40]. Gangliogliomas (GG) and dysembryoplastic neuroepithelial tumours (DNET) are the most common LEATs [39,40].

Signs of cortical dyslamination and hypoplastic changes (loss of the six-layer structure of the cortex and/or heterotrophic neurones) may be seen in close proximity to a GG, DNET, or another neoplasm associated with epilepsy [41]. It is sometimes a real challenge to differentiate alterations associated with tumour infiltration from cortical dyslamination in the context of FCD type IIIb.

### 6.1. Gangliocytoma, Ganglioglioma, Anaplastic Ganglioglioma, and Desmoplastic Ganglioglioma

Both GG and gangliocytomas (GC) are low-grade tumours, composed of mature yet cytologically atypical neuronal cells alone (GC) or in combination with a well-differentiated glial component (GG). These tumours are slow-growing and often associated with chronic temporal lobe epilepsy [7]. The age at diagnosis varies greatly—from 2 months to 70 years. The majority of GG cases (>70%) affect the temporal lobe. Other intra-axial brain localisations in descending order of frequency are frontal, parietal, and occipital lobe [40,42].

A common MRI finding is a “cyst with mural nodule”, which is also seen in other low-grade brain tumours—pilocytic astrocytoma and pleomorphic xanthoastrocytoma [43].

GC are composed of disorganised atypical, but well-differentiated ganglion cells. Neurones with large vesicular nuclei and centrally located prominent nucleoli, abundant cytoplasm, abnormal Nissl substance, and multipolar protrusions are also found [44].

Macroscopically, GG are usually dense, well-demarcated formations with a granular cut surface. Histopathologically, GG are made out of proliferated glial cells and neurones, and occasionally contain bi- or multinuclear atypical neurones (Figure 4A) [45]. Although the two conditions are identical in terms of the neuronal component, GG is diagnosed much more often than GC. The predominant glial cells found in GG are astrocytes. Frequent findings in GG are perivascular and/or interstitial lymphocytes and plasma cell infiltrates. Less often present in GG are stromal calcification, fibrous thickening of vessel walls, eosinophilic granular bodies, and Rosenthal fibres. Cortical dyslamination is a common pathomorphological finding [44]. The neural component can be immunohistochemically verified with various markers such as synaptophysin (Figure 4C), neurofilament protein, MAP2 (Figure 4D), NeuN, and NSE [46]. The glial cell component can be revealed by a positive reaction for GFAP, Olig-2, and S-100 protein. The proliferative index (Ki-67) of conventional GG is low (<5%) and usually limited to the glial component of the tumour [47].

Conventional GG corresponds to grade 1 according to the WHO classification of CNS tumours [48]. In terms of differential diagnosis, pleomorphic xanthoastrocytoma should be considered, which is distinguishable from GG due to the presence of larger cells positive for GFAP exhibiting glial differentiation. In contrast, GG is characterised by atypical neuronal cell component, which is GFAP-negative but synaptophysin-positive. DNET is another tumour that exhibits mature, so-called “floating” neurones and a low-grade glioneuronal component with distinctive nodular character, microcystic zones, and oligodendrocyte-like cells. Low-grade gliomas sometimes contain neurones infiltrated by neoplastic cells, where secondary structures such as perineuronal satellitosis are detected. Diffuse glioma is definitively diagnosed by establishing IDH mutations.

The majority of GG (>80%) contain ramified cells either within the tumour or in the adjacent cortex that express the oncofoetal epitope CD34, which is not normally expressed outside of vascular endothelial cells in the mature brain (Figure 4B) [49]. For cases with diagnostic uncertainty, the initial molecular workup should focus on BRAF p.V600E mutation testing either by sequencing or by IHC using the mutant-specific antibody VE1 [50].

Conventional GG and GC are well-differentiated, clinically indolent tumours with excellent prognosis [44]. Following tumour surgical resection, 88% of patients with pharmacoresistant epilepsy were seizure-free during a 7-year follow-up. Tumour recurrence is detected in less than 2% of cases [51,52].

Anaplastic ganglioglioma (AGG) is a glioneuronal tumour composed of dysplastic neurones and an anaplastic glial component with increased mitotic activity [48]. AGG corresponds to the WHO’s grade 3 tumour [52].

The anaplastic changes in AGG predominantly affect the glial cells (Figure 4E,F) [48]. Tumours with malignant transformation of neuronal cells are rarely reported [53]. Although the condition’s aetiology and pathogenesis remain elusive, it has been hypothesised that AGG originates from glioneuronal precursor cells [54,55]. Anaplastic features typically appear de novo, predominantly in paediatric cases [56]. Albeit a rare occurrence, it is possible for the anaplastic transformation to originate from benign ganglioglioma, particularly in older patients [57]. Park et al. (2002) determined that the two factors associated with best prognosis were low tumour grade and complete surgical resection of the lesion [58]. Although there is inconclusive evidence on the effectiveness of radiotherapy in GG patients, this approach is adopted in cases with cancer recurrence, where total tumour resection is impossible, or in neoplasms with anaplastic features or containing oligodendroglial-like cells [58,59].

A rare GG variant is desmoplastic infantile ganglioglioma (DIGG). It is characterised by desmoplastic stroma and mature neuronal cell component (Figure 4G,H). DIGG affects patients in early childhood and is typically diagnosed during the first two years of life (mean age at diagnosis is 6 months) [48]. Characteristic histological features are collagen- and reticulin-rich areas with cluster-forming spindle-shaped cells, occasionally exhibiting a so-called “storiform pattern” [44]. DIGGs are typically supratentorial lesions, clinically manifesting with hypertonus, tense fontanelles with rapid increase in head circumference. Epileptic seizures and pareses can also be observed. DIGG corresponds to the WHO’s grade 1 tumour. Tumour recurrence has not been described during long-term follow-up [60].

### 6.2. Dysembryoplastic Neuroepithelial Tumour

Dysembryoplastic neuroepithelial tumour (DNET), first described in 1988 by Daumas-Duport et al. [61], is a slow-growing tumour, histologically corresponding to the WHO’s grade 1 tumour [48]. It is usually diagnosed in childhood and is commonly associated with pharmacoresistant epilepsy. It is localised in the cerebral cortex, primarily in the mesial part of the temporal lobe, and less often it is found frontally [62]. Much less frequent, it can be found in other areas such as the septum pellucidum, basal nuclei, brainstem, cerebellum, and extremely rarely it is found intraventricularly, in the corpus callosum or in its surrounding tissues [63,64,65].

According to the literature, in patients with recurrent unprovoked epileptic seizures, in the context of pharmacoresistant epilepsy, DNET represents 18% of tumours in adulthood and 23% of neoplasms in children [66].

Neuroradiological findings are important in distinguishing DNET from tumour glial proliferations, especially oligodendrogliomas. Most DNETs are well-defined formations in the cerebral cortex. MRI imaging demonstrates a T1-hypointense lesion with an intratumoural nodular character. T2-weighted imaging reveals a well-demarcated hyperintense lesion, usually found in the temporal lobe. The majority of DNETs are non-contrasting formations, they are very rarely contrast, and extremely rarely even show peripheral (ring-) contrast [67,68].

Pathomorphologically, DNET can be divided into “simple” and “complex” histological variants. Common to both forms are the presence of specific glioneuronal elements—oligodendrocyte-like cellular elements, “floating” mature neurones without dysplastic changes, and intracortical, mucin-like spaces with microcystic characteristics (the latter give a positive histochemical reaction for acidic glycosaminoglycans when examined with Alcian blue, see Figure 5B) [44].

The so-called simple histological forms of DNET include distinctive pathomorphological features: specific glioneuronal complexes organised in microcolumns of small, round, oligodendrocyte-like cells arranged along the course of axonal processes or microcirculatory blood vessels, perpendicularly oriented to the pial surface. The described glioneuronal complexes are separated by a myxoid or mucin-like substance-rich matrix. Among microcystic spaces of different sizes, floating mature neurones without dysplastic changes are seen. Stellate astrocytes are sometimes also found. The complex histological variants of DNET are characterised by the appearance of glial nodules (Figure 5A), which can be histologically indistinguishable from pilocytic astrocytoma or oligodendroglioma. These glial nodules are sometimes associated with the presence of hamartomatous, malformative, and rarely, calcified blood vessels. In the literature, there are also single cases with a combination of DNET and GG [44,69]. Both histological variants of DNET can be associated with FCD found in the internodular spaces or in the cerebral cortex in close proximity. Mitotic figures are usually absent or extremely rare. Necrotic fields are also observed exceptionally, and their presence is not always associated with a worse prognosis for the patient [44].

Distinguishing DNET from oligodendroglioma can become a challenge, especially on small and fragmented biopsy materials. The nodular character, the presence of glioneuronal elements, as well as the appearance of floating neurones in the absence of perineuronal satellitosis, favour DNET. In adult patients, an additional molecular genetic analysis is recommended—the presence of IDH mutation or chromosomal 1p/19q codeletion confirms the diagnosis of oligodendroglioma and, accordingly, rejects DNET. However, in the paediatric group of patients, the lack of IDH- mutation and 1p/19q codeletion cannot completely exclude oligodendroglioma. DNET and oligodendrogliomas diagnosed in children can demonstrate FGFR1-alterations, which point to a possible biological correlation between the two tumours [70].

DNETs are complex lesions made up of glial and neuronal cells. Stellate and piloid cells with astroglial differentiation are immunohistochemically positive for GFAP, while mature neuronal cells give a positive immunohistochemical reaction for NSE, neurofilament, NeuN, MAP2, and synaptophysin [71]. Oligodendrocyte-like cells are immunohistochemically positive for Olig-2 and S-100 protein (Figure 5C). The Ki-67 proliferative index demonstrates very low values (Figure 5D)—usually 1–2%, but cases with up to 8% proliferative activity have also been reported [72].

Classic histological variants of DNET have a poorly expressed growth potential, and complete resection of the tumour usually results in complete control of seizures. In a case series with DNET, 85% of the children were seizure-free at the 1-year follow-up and 65% at a longer time interval; there was an average of 4.3 years of follow-up [73]. Elderly patients and those with a longer duration of epilepsy are at higher risk of relapses. However, cases of tumour recurrence after subtotal or even apparent total removal of the lesion have rarely been reported [74]. Currently, the noteworthy histologic features of DNET are increased number of mitotic figures, microvascular proliferation, and areas of necrosis with pseudopalisading arrangements. None of these have been fully confirmed to correlate with a worse patient prognosis. It is worth noting that the so-called “malignant transformation” has been observed in individual cases, which does not allow making definitive conclusions about their biological behaviour [44].

### 6.3. Pleomorphic Xanthoastrocytoma

Pleomorphic xanthoastrocytoma (PXA) is a rare neoplasm, accounting for less than 1% of all CNS tumours. It is found in children and young individuals, and is often associated with focal epilepsy, but with a relatively good prognosis [75,76].

PXA is primarily diagnosed during the first two decades of life. Most often, it develops in the cortex of the cerebral hemispheres, particularly in the temporal lobe, and invades the overlying leptomeninges [76].

Imaging studies (CT, MRI) show a solitary supratentorial mass, often involving the leptomeninges. Cystic and contrast-enhancing lesions are common. The classic tumour configuration is demonstrated by a cyst and a contrasting mural nodule [77].

Macroscopically, PXAs are well-demarcated formations, usually with a yellowish cut surface, which is explained by the lipidised astrocyte cellular composition. A cystic component is present, sometimes with calcifications [44].

Histologically, large pleomorphic cells with astrocytic- or mesenchymal-like morphology are found. Bizarre nuclei with multinucleated cell shapes and abundant eosinophilic cytoplasm are characteristic. Smaller polygonal or spindle-shaped cells are also found, often forming bundles [78]. Important for the diagnosis of PXA are the xanthoma cells (“xanthoastrocytes”) with their foamy cytoplasm (Figure 5E). Another important morphological characteristic is the low number of mitotic figures, despite the well-known pleomorphic appearance of the neoplasm. Tumour cells are separated by reticular fibres, the intercellular reticulin, which help distinguish PXA from other brain neoplasms containing pleomorphic cellular elements. An essential, although not specific, finding is the appearance of eosinophilic granular bodies, which are usually absent in high-grade gliomas. Areas of necrosis and presence of microvascular proliferation are morphologic features that occur in the malignant transformation of PXA. When these manifestations are combined with more than 5 mitotic figures/10 HPF, the diagnosis of anaplastic PXA is made (Figure 5F) [44].

Conventional PXA corresponds to second grade, while anaplastic forms correspond to third grade tumour classification according to the WHO [48]. An interesting and controversial fact is that PXAs with malignant transformation lose a substantial part of the pleomorphic cellular composition and acquire a more “monomorphic” appearance [44].

Differential diagnosis is made with giant cell glioblastoma, gliosarcoma, epithelioid glioblastoma, and pleomorphic sarcoma. An important point here is the proof of various genetic aberrations—IDH mutations, gain of chromosome 7, EGFR amplifications, and loss of chromosome 10, which are typical for glioblastoma or high-grade astrocytomas. BRAF mutations are common for both PXA and epithelioid glioblastoma, but the latter is immunohistochemically negative for intracellular reticulin [44,48].

The Ki-67 proliferative index is usually less than 3% in conventional PXA, corresponding to the WHO’s grade 2, while more than 10% proliferative activity points to anaplastic PXA, which are WHO grade 3 tumours [44].

PXA is associated with a good prognosis, as in the majority of the cases complete surgical removal is the definitive treatment. In anaplastic PXAs, as well as in cases with partially removed conventional PXAs, radio- and chemotherapy are considered. In the rare cases of proven MGMT promoter methylation, treatment with temozolomide is also introduced [79], and in proven BRAF mutations, targeted therapy with vemurafenib can be applied [80].

### 6.4. Rosette-Forming Glioneuronal Tumour

Rosette-forming glioneuronal tumour (RFGNT) is a rare, slow-growing WHO first grade tumour affecting young individuals. It develops in the posterior fossa, with involvement of the fourth ventricle. The tumour manifests clinically much more often with headache and ataxia than with seizures. The average age at diagnosis is 27 years, and it is slightly more common in women [81].

MRI reveals a circumscribed, solid mass, sometimes with a cystic component, which often shows peripheral contrast [82].

Histologically, RFGNT has biphasic nature. The neurocytic component contains a monomorphic population of small cells with eosinophilic cytoplasm, forming rosettes around eosinophilic neuropil, as well as pseudorosettes formed around small-calibre blood vessels (Figure 6A). Neurocytic cell elements are immunohistochemically positive for synaptophysin (Figure 6B). The glial component is solid and represented by cells with a piloid morphology, with occasional oligodendrocyte-like round cells, microcystic spaces, Rosenthal fibres, and eosinophilic granular bodies. Both components bear the hallmarks of a low-grade brain tumour [83].

The complete surgical removal of the tumour is evidenced to be its definitive treatment. In the literature, there are single reports of malignant transformation with dissemination in the CNS [84].

### 6.5. Papillary Glioneuronal Tumour

Papillary glioneuronal tumour (PGNT) is a low-grade neoplasm, WHO grade 1 [48] composed of glial and neuronal (predominantly neurocytic) cells presenting papillary configurations [85].

PGNTs are extremely rare, accounting for less than 0.02% of all CNS tumours. The mean age at diagnosis is 23 years [86].

The clinical manifestation of PGNT includes epileptic seizures, headaches, visual disturbances, etc. The majority of PGNTs are supratentorial lesions, primarily involving the temporal lobe. Imaging data are non-specific [44].

A definitive diagnosis is made only after histological examination, which is illustrated by pseudopapillary structures containing hyalinised vessel walls surrounded by a biphasic neoplastic cell population. Immediately adjacent to the fibrovascular axes (hyalinised vessel walls) are spindle-shaped to cuboidal glial cells that are positive for GFAP by immunohistochemistry. NeuN- and synaptophysin-positive neuronal cells are found between the pseudopapillary structures. Very rarely, PGNTs show signs of transformation to a high-grade tumour such as increased mitotic activity (with the appearance of atypical mitotic figures), manifestations of vascular proliferation, and areas of necrosis [87].

Differential diagnosis is made between extraventricular neurocytoma, astroblastoma, ependymoma, and RFGNT. A major distinguishable feature of PGNT is its characteristic immunohistochemical profile, verifying the condition’s biphasic nature. In addition, a specific translocation has been demonstrated in the tumour—t (9;17) (q31;q24) [88].

A case series on 71 patients pathomorphologically diagnosed with PGNTs demonstrated a real possibility of total surgical resection correlating with definitive treatment of the patients [87].

### 6.6. Hypothalamic Hamartoma

Hypothalamic hamartoma (HH) is a tumour-like lesion with minimal growth potential, developing in the hypothalamus tuber cinereum or from the floor of the third ventricle. Although HH can be asymptomatic, in some cases, there are manifestations in the form of epileptic seizures with sudden onset of uncontrollable laughter (gelastic epilepsy) or premature puberty [89]. Its frequency varies widely according to different authors. Weissenberg et al. (2001) set its incidence at 1 in 50,000–100,000, while Ng et al. (2005) estimate that it is diagnosed in as few as 1 in 1,000,000 [90,91].

Macroscopically, HH resemble the grey matter of the brain. Histologically, they are represented by collections of mature, small neurones and glia (Figure 6C,D). The neuronal component is usually dominant, with a partly nodular character. The nodes vary in size—some of them contain up to ten, and others contain more than a hundred neurones, which are partially separated by a hypocellular neuropil. Less commonly, the dominant cellular components are well-differentiated glial cells. In these cases, fibrillar astrocytes and, more rarely, oligodendrocytes and microglial cells are observed. Regarding Ki-67, the proliferative index is close to 0%, with individual positive nuclei of reactive cells, which once again points to the hamartomatous nature of the lesion [44,92].

Currently, the choice of treatment for HH is their disconnection rather than their complete surgical removal due to the presence of significant risks associated with their resection [93].

### 6.7. Angiocentric Glioma

Angiocentric glioma is a WHO grade 1, slow-growing supratentorial tumour observed in children that is strongly associated with seizures [48]. The incidence of angiocentric glioma is extremely low. The average age at diagnosis is 16 years. It is predominantly found in the temporal, frontal, or parietal lobe. Almost all cases manifest clinically with long-lasting pharmacoresistant epilepsy [94].

Histologically, angiocentric glioma is composed of monomorphic bipolar tumour cells. They are spindle-shaped, with oval or elongated nuclei and eosinophilic cytoplasm. There is a tendency for perpendicular arrangement of tumour cells around the pia mater and vessel walls. Similarly to other low-grade epilepsy-associated tumours, hallmarks of FCD may be seen adjacent to angiocentric gliomas [95].

Differential diagnosis is made with ependymoma, pilocytic astrocytoma, and low-grade astrocytoma. The first two tumours show a tendency to form a tumour mass, and the third demonstrates a diffuse growth, similarly to the angiocentric glioma, but without a tendency for perivasal arrangement of proliferating glial cells [44].

Tumour cells in angiocentric glioma show positive immunohistochemical reaction for GFAP, S-100 proteins, and vimentin, and exhibit a “dot-like” cytoplasmic reaction for EMA. The proliferative index (Ki-67) is low, ranging from 1% to 5% [44,96].

Complete surgical removal of these tumours leads to complete seizure control. There is a case description of a patient with initially histologically verified low-grade angiocentric glioma, who relapsed after 21 months, exhibited morphological signs of anaplastic transformation, and subsequently suffered a lethal outcome [94,97].

### 6.8. Isomorphic Diffuse Astrocytoma

Isomorphic diffuse astrocytoma (IDA) was first described in 2004 in patients with early-onset pharmacoresistant focal epilepsy corresponding to the first degree tumours of the WHO [98,99]. To date, the largest IDA case series includes 20 patients with a mean age of 29 years, a male predilection, and proven MYB and MYBL1 genetic alterations [100]. IDAs are most often localised in the temporal lobe. Microscopically, they are composed of diffusely proliferated iso/monomorphic glial cells in a fibrillary matrix. It is worth noting that tumour cells are very difficult to distinguish from normal brain parenchyma. They are positive for GFAP and negative for Olig2, CD34, and IDH1p.R132H. The proliferative index determined with the Ki-67 marker is very low [98,100].

## 7. Epileptogenic Vascular Malformations, including in the Context of FCD IIIc, Meningioangiomatosis

### 7.1. Cavernous Hemangioma (Cavernoma)

Cavernomas are vascular malformations with nodular character composed of cavernous spaces lined with endothelial cells but lacking the typical histological structure of “mature” blood vessel wall (Figure 7A). MRI reveals well-demarcated surface cortical lesions with adjacent hemosiderin deposits. Histological examination can reveal calcifications and sometimes even ossification. The incidence of cavernomas varies between 0.02 and 0.5% [101,102].

Around 4–6% of cases with pharmacoresistant epilepsy are caused by cavernous vascular malformations. The seizures are typically associated with blood products that induce reactive perilesional gliosis. The recurrent microhaemorrhages, and consequently the hemosiderin deposits in the cortical tissue, likely cause hyperexcitability through ferrous ions and subsequent oxidative damage and lipid peroxide formation causing proliferation of epileptogenic glial cells [102].

### 7.2. Arteriovenous Malformation (AVM)

AVMs can be found in all parts of the CNS; however, they predominantly affect the middle cerebral artery pool. MRI findings include a limited lesion with hemosiderin deposits. Angiography reveals arteriovenous shunting. Histological examination allows visualisation of vessels with different calibre and thickness of the tunica media. Typical features in the vicinity of the lesion include signs of gliosis and clusters of hemosiderophages, which are indicative of recurrent haemorrhages. Although AVMs can manifest at all stages of life, most of them present between the ages of 20 and 40. This type of malformation “steals” a significant portion of the blood supply from adjacent brain regions, which may be the underlying cause of seizures in patients with AVM. Hemosiderin deposits may also contribute to epileptogenesis [103].

### 7.3. Sturge–Weber Syndrome

Focal epilepsy has been reported in around 80% of Sturge–Weber syndrome patients. The prevalence of the condition is about 1/50,000. Its inheritance pattern remains elusive [104].

Typical diagnostic imaging and macroscopic findings include extensive unilateral leptomeningeal angioma, cortical “railway” calcifications, and less frequently atrophy of the subjacent neocortex. The white matter presents signs of chronic ischemia and reactive gliosis. Functional MRI examination reveals reduced glucose metabolic activity and hypoperfusion of the affected cortical region. Microscopically, large curved venous structures among thickened leptomeninges (resembling leptomeningeal angioma) are found (Figure 7B,C). Signs of atrophy and neuronal loss, reactive astrogliosis, and microcalcification are found in the subjacent brain parenchyma. Another typical finding is an increased number of cerebral vessels with thickened walls as a result of hyalinisation and subendothelial proliferation [104].

The epileptogenic substrate in Sturge–Weber syndrome is cellular damage caused by excessive vascular proliferation which disrupts the normal blood supply to the brain tissue and leads to atrophy and focal calcification of the adjacent brain parenchyma [105].

### 7.4. Meningioangiomatisis

Meningioangiomatosis (MA) is a rare tumour-like meningovascular hamartomatous lesion [106]. The majority of cases with MA are sporadic, although it may alsooccur in association with neurofibromatosis type 2 (NF2). The lesions are histologically identical, but differ in their clinical manifestation. Sporadic cases are usually symptomatic, almost exclusively with seizures, although NF2-associated lesions are most commonly detected as incidental findings [107]. MA occurs in patients of all ages and is most commonly identified in the temporal lobe, followed by the frontal, parietal, and occipital lobes in descending order of frequency [108]. The pathognomonic finding of this lesion is its invasive nature, characterised histologically by meningovascular proliferation, perivascular cuffs of spindle cells, and neurofibrillary tangles (Figure 7D) [107,108]. The meningovascular proliferation and the spindle cells are IHC negative for S-100 protein and progesterone receptor (usually). CD34 highlights the vascular elements and the MIB-1 index is very low [109].

The MA is benign and surgically treatable, although resection can be complicated by the invasive character of the lesion [107,110,111].

FCD type IIIc is histopathologically defined by the ILAE Classification, 2022, as abnormal cortical organisation adjacent to epilepsy-associated vascular malformations [4]. In a retrospective series of surgical brain specimens from 14 epilepsy patients with leptomeningeal angiomatosis in Sturge–Weber syndrome, and an AVM, FCDIIIc was histopathologically confirmed in all patients with Sturge–Weber syndrome and can be best described as cortical pseudolaminar sclerosis—horizontally organised layer abnormalities, including neuronal cell loss and astrogliosis [112].

## 8. Epileptogenic Encephalitides, Focal Cortical Dysplasia in Combination with Another Epileptogenic Lesion, Acquired Earlier in Life (Traumatic Injuries, Ischemic Injuries and Encephalitis)—FCD Type IIId

Epileptic seizures are commonly observed in encephalitis patients. Cytotoxic T-lymphocytes and antibody-mediated complement activation are the most important components of the adaptive immune system, inducing neurodegenerative changes that can subsequently lead to epileptogenic encephalitis. On the other hand, the innate immune system acts through interleukin-1 and Toll-like receptor-associated mechanisms, playing a direct role in epileptogenesis [113,114,115].

In 1958, the Canadian neurosurgeon Theodore Rasmussen published a scientific report entitled: “Focal seizures due to chronic localised encephalitis” in three childhood patients. Rasmussen’s encephalitis involves one hemisphere. It begins with a “prodromal period” that sometimes lasts for years, during which mild hemiparesis or rare seizures may be observed. This is followed by the “acute phase” of the disease, characterised by more frequent pharmacoresistant unilateral simple partial seizures or secondary generalised seizures. Over the course of the disease, the inflammatory manifestations spread through the hemisphere, which determines the appearance of new epileptogenic zones. A few months after the onset of the epileptic seizures, neurological deficits in one hemisphere follow, including hemiparesis, hemianopia, impairment of cognitive function, and aphasia [116]. The neuropathological hallmarks of Rasmussen encephalitis are inflammatory manifestations—perivasal lymphocytic infiltrates in the cerebral cortex and underlying white matter, neuronal loss, microglial activation, microglial nodules, and astrogliosis (Figure 8). There is a focus on the presence of antibodies against glutamate receptor 3 (GluR3) found in the serum of patients with Rasmussen’s encephalitis in scientific reports from the 1990s. These antibodies are thought to lead to an antigen- or complement-mediated attack on neurones. In addition, cytotoxic T-lymphocytes are thought to be responsible for the neurodegeneration and astrocyte loss in Rasmussen’s encephalitis, which plays an important role in epileptogenesis [117,118,119,120].

The second group of encephalitis in which epileptic seizures are observed are antibody-associated; these are classified as paraneoplastic and non-paraneoplastic limbic encephalitis [80]. Many patients with paraneoplastic encephalitis have serum antibodies reactive against both the primary neoplastic process and nervous system structures [121,122]. A role for cytotoxic T-lymphocytes is best documented in “classic” paraneoplastic encephalitis with formed antibodies against oncogenes. Moreover, CD8-positive T cells are the dominant cellular component in neuropathological studies of paraneoplastic encephalitis with the presence of anti-Hu, anti-Yo, or anti-Ma antibodies [80].

In the limbic encephalitis, histologically in the temporal neocortex and in the hippocampus, there are pronounced signs of astrogliosis, microglial activation with a tendency to form nodules, and infiltration of T-lymphocytes in the brain parenchyma [121,122].

After a longer period of time, the only biopsy finding in patients with encephalitis may be marked “cellular” gliosis.

FCD IIId is most prominent in regions of pre- or perinatal stroke and adjacent to the resulting porencephalic cyst. Another clinico-pathological FCD IIId subtype is that of neocortical layer 4 loss in the occipital lobe associated with perinatal hypoxemic disturbances occurring predominately in male children [123]. FCD IIId may also occur in early onset Rasmussen encephalitis [124] or traumatic brain injuries with reactive gliosis acquired during early life [4].

## 9. Epileptogenic Lesions with Predilection for White Matter Involvement

According to the revised International League Against Epilepsy (ILAE) classification 2022 of FCD, in cases with the presence of heterotopic neurones in the underlying cerebral white matter, and in the absence of other structural lesions, the condition is defined as a mild malformation of cortical development (mMCD) [4]. The diagnosis of mMCD requires the detection of more than 30 neurones/mm^2^ in the cerebral white matter using the IHC markers MAP2 or NeuN [4].

MOGHE is diagnosed by the presence of characteristic histological findings—an increased number of heterotopic neurones in the underlying cerebral white matter (>30 neurones/mm^2^) and an increased oligodendroglial density with blurring of the grey–white matter boundary (over 2000 Olig2 immunoreactive cells per mm^2^) (Figure 9) [5]. Another structural lesion from the spectrum of FCD needs to be excluded. The described cases primarily affect the frontal lobe at an early age, with a typical onset of epilepsy usually at 2 years of age (ranging from 0.3 to 13 years) [5].

SLC35A2 somatic mutations are demonstrated in 45–100% of MOGHE patients [125,126]. Bonduelle et al. (2021) demonstrated somatic pathogenetic SLC35A2 variants in 9 of a total of 20 molecularly genetically examined cases [126]. The gene encodes a UDP-galactose transporter in the membranes of the Golgi apparatus. The establishment of SLC35A mutations in the brain tissue of patients with pharmacoresistant epilepsy provides a new opportunity for a personalised non-invasive therapeutic approach with oral galactose supplementation. This treatment strategy would be particularly valuable in cases where total surgical resection of the epileptogenic lesion is impossible, for instance when there are multiple foci or when the epileptogenic zone affects extensive or critical brain regions [126].

In 2 to 26% of patients operated on due to pharmacoresistant epilepsy, the pathomorphological examination does not allow a definite histological diagnosis (“no FCD on histopathology”). In some of these patients, preoperative imaging showed an abnormal finding that was not confirmed morphologically [4,5].

## 10. “Dual” and “Double Pathology”

Principal epileptogenic lesions comprise any morphological lesion with aetiologically defined pathogenesis of either neoplastic, genetic, infectious, traumatic, or metabolic origin. Principal lesions include different epilepsy-associated tumours, vascular malformations, encephalitis, malformations of cortical development, traumatic scars, etc. [7].

“Dual Pathology” refers to patients with HS, who have a second principal lesion other than FCD type I, i.e., tumour, glial scar, encephalitis, vascular malformation, FCD type IIa, or FDC type IIb [7]. Histologically confirmed architectural abnormalities in the temporal lobe associated with HS should not be diagnosed as FCD type I or dual pathology but FCD type IIIa [7].

“Double Pathology” refers to two independent lesions affecting one or multiple lobes, but not including HS. Both lesions evolve from an independent pathogenesis, e.g., cavernous hemangioma in one cerebral hemisphere and a GG in the other [7].

The complete surgical resection of all the various principal lesions is crucial in cases with Dual and Double Pathology as each lesion itself represents an epileptogenic focus.

## 11. The Integrated Approach for the Diagnosis of FCD

Targeted imaging and molecular genetic studies provide important clues in the diagnosis of different histopathological variant of FCD. 

MRI data is an essential cornerstone in the workup of patients with focal epilepsy [127]. Certain MRI findings could be predictive of the FCD type, e.g. presence of a ‘transmantle sign’ in FCD IIb [128]. It is noteworthy that MRI could be negative in some histopathological confirmed FCD IIa or in cases with mMCD, and MOGHE. Ultra-high field MRI could further improve the diagnostic yield in such cases [129,130]. 

Genetic testing of somatic and germline mutations for FCD is currently not available in most pathology laboratories. It is an important diagnostic tool for determining whether patients with FCD carry pathogenetic variants. Different genes (AKT3, DEPDC5, MTOR, TSC1, TSC2, others) have been reported to cause canonical FCD IIa [125]. SLC35A2 should be included in the panel in order to differentiate MOGHE [126]. The diagnostic yield using such gene panel sequencing from routine formalin-fixed paraffin-embedded or frozen brain tissue ranges from 32% when assessing various epilepsy associated lesions [125], to 45% in patients selected for MOGHE [126], and 63% in patients with FCD type II [125].

The genetic panel needs to be expanded, especially in histopathologically proven cases of FCD type Ib and Ic, where the exact aberrations are not yet known. The leptomeningeal vascular malformations also represent an area where more genetic studies are needed.

## 12. Conclusions

In summary, the diverse spectrum of FCD includes isolated forms (FCD type I—characterised by histoarchitectonic alterations of the cerebral cortex, and type II—defined by the presence of typical dysmorphic neurons and balloon cells), forms associated with other principal epileptogenic lesions (FCD type III), as well as cases with predilection for white matter involvement (mMCD, MOGHE). Although the pathomorphological characteristics of individual forms are well known, their aetiological and pathogenetic features are not fully understood and need to be explored further. 

The adult group of patients with medically refractory epilepsy is dominated by HS. In the pediatric group most cases present with isolated forms of FCD—types I and II. Glioneuronal tumours are a very common finding in both groups. The most common long-term epilepsy associated tumours are GG and DNET which are typically benign, low-grade tumours. Malignant transformation may be observed exceedingly rarely. 

Additional immunohistochemistry analyses prove invaluable in the diagnosis of FCD. NeuN staining allows visualisation of the abnormal histoarchitectronics of the neocortex in FCD type I cases. In addition, it enables the detection of heterotropic neurones which are a defining feature of mMCD and MOGHE. Of particular utility for the diagnosis of FCD type II are the markers SMI32 Neurofilament (staining dysmorphic neurones in FCD IIa) and vimentin (revealing the balloon cells in FCD IIb). The neuronal loss and fibrillary gliosis in HS can be proved with the IHC markers NeuN and GFAP, respectively. Useful markers in cases with LEAT are CD34 (for GG cases), Alcian blue (revealing the nodules in DNET cases), Reticulin (illustrating increased number of reticular fibres in PXA and DIGG). Although high-grade transformation of LEAT is extremely rare, it can be IHC verified by Ki-67 revealing increased proliferation index.

In terms of management of pharmacoresistant epilepsy, targeted MRI investigation of epileptogenic lesions, their complete surgical removal and correct histological diagnosis seem to be the most effective approach. Moreover, imaging and genetic examination may provide further insight into the mechanisms underlying these conditions thereby opening novel therapeutic avenues as has been the case with FCD type II (mTOR inhibitors) and MOGHE (oral galactose supplementation).

## Figures and Tables

**Figure 1 diagnostics-13-01311-f001:**
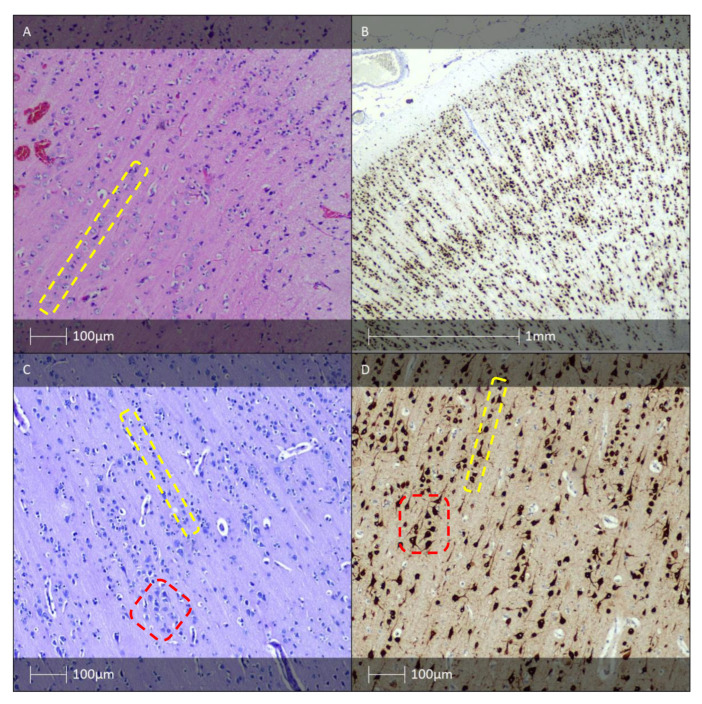
FCD type I. (**A**) FCD type Ia—Hematoxylin and eosin (H&E) staining with vertical organisation of pyramidal cells (yellow dotted line). (**B**) Microcolumnar arrangement of the neocortex (NeuN IHC). (**C**) FCD Ic pattern—H&E staining with vertical (yellow dotted line) and horizontal (red dotted line) cortical dyslamination of the neocortex. (**D**) FCD Ic pattern—vertical (yellow dotted line) and horizontal (red dotted line) cortical dyslamination of the neocortex (NeuN IHC).

**Figure 2 diagnostics-13-01311-f002:**
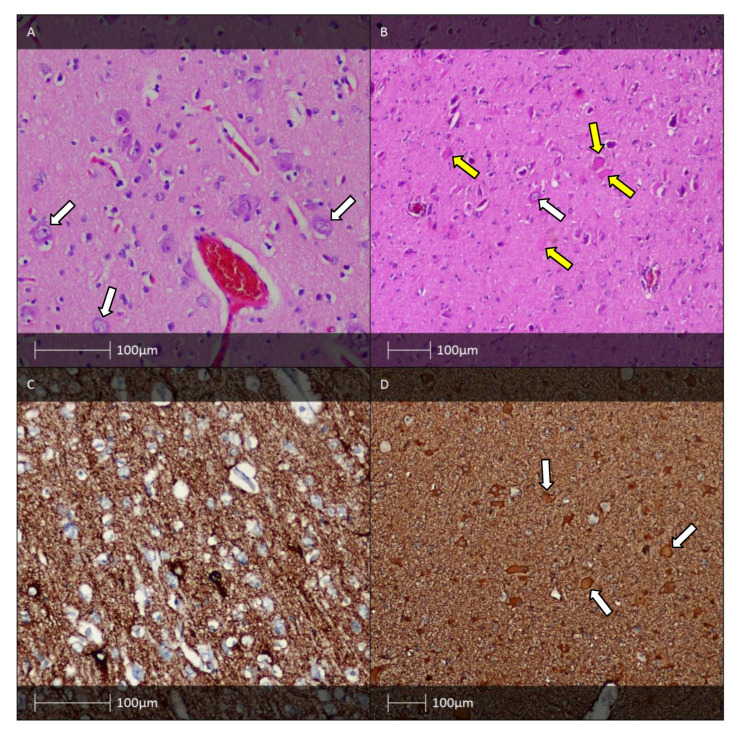
FCD type II. (**A**) FCD type IIa—H&E staining showing dysmorphic neurones with enlarged cell bodies as well as enlarged nuclei, and abnormal Nissl substance (white arrows). (**B**) FCD IIb—H&E staining with dysmorphic neurone (white arrow) and balloon cells (yellow arrows). (**C**) The dysmorphic neurones with prominent accumulation of neurofilament protein (SMI32 IHC). (**D**) FCD IIb pattern—vimentin immunohistochemistry labels the balloon cells (white arrows, Vimentin IHC).

**Figure 3 diagnostics-13-01311-f003:**
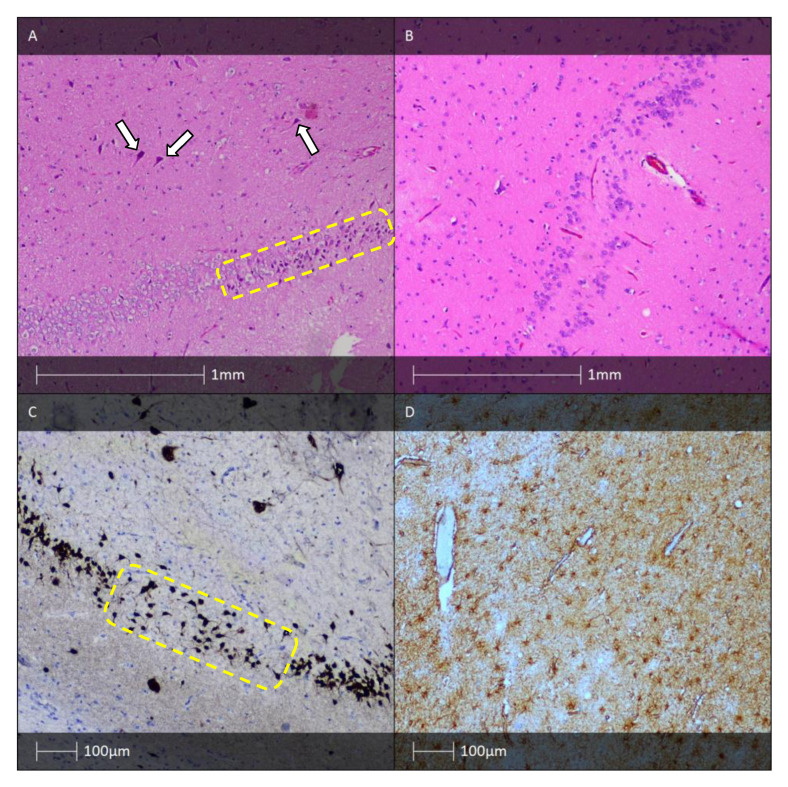
HS. (**A**) HS—H&E staining showing fibrillary astrogliosis (white arrows) and neuronal loss (yellow dotted line). (**B**) HS—The granule cell layer with a bi-laminated architecture (H&E). (**C**) HS—pyramidal cell loss (NeuN IHC). (**D**) HS—fibrillary astrogliosis (GFAP IHC).

**Figure 4 diagnostics-13-01311-f004:**
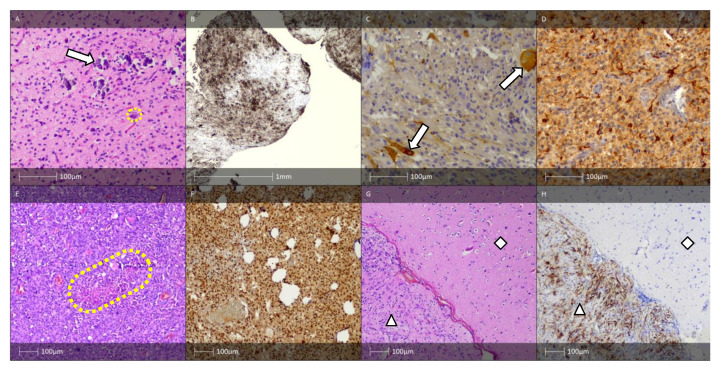
GG, AGG, and DIGG. (**A**) GG—H&E staining showing an astroglial component with small, round to oval nuclei with open chromatin structure and microcalcifications (white arrow). Dysplastic neurones are also visible (yellow dotted line). (**B**) Areas of CD34 immunopositive tumour cells and tumour cell satellites (CD34 IHC). (**C**) Synaptophysin-positive dysplastic neurones (white arrows, synaptophysin IHC). (**D**) MAP2 immunopositive tumour cells (MAP2 IHC). (**E**) AGG—H&E staining showed a tumour with increased cellularity and necrotic changes (yellow dotted line). The majority of tumour cells show rhabdoid features. (**F**) Anaplastic ganglioglioma—rhabdoid cells show a preserved nuclear expression of the marker INI1 (IHC). (**G**) DIGG—H&E staining showing tumour component (white triangle) and sharply demarcated neocortex (white rhombus) (HE). (**H**) DIGG—areas of CD34 immunopositive tumour cells (white triangle) and negative reaction in the sharply demarcated neocortex (white rhombus) (CD34 IHC).

**Figure 5 diagnostics-13-01311-f005:**
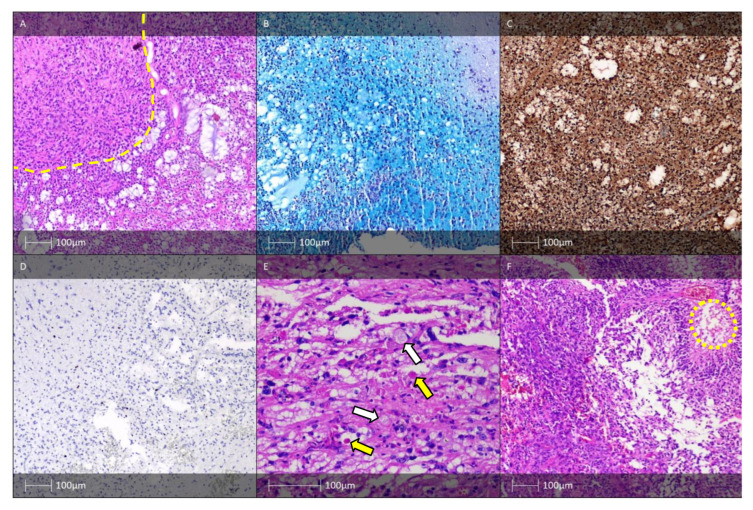
DNET, PXA, and APXA. (**A**) DNET—glial nodule (yellow dotted line), and characteristic glioneuronal complexes (HE). (**B**) DNET, histochemical examination—Alcian blue. (**C**) DNET—IHC—S-100-protein positivity in oligodendrocytic-like cells. (**D**) DNET—low proliferative index, Ki-67 IHC. (**E**) PXA—“xanthoastrocytes” with foam cytoplasm (white arrows) and eosinophilic globular bodies (yellow arrows). (**F**) Anaplastic PXA—a high-grade tumour with palisading cells around a central focus of necrosis (yellow dotted line), HE.

**Figure 6 diagnostics-13-01311-f006:**
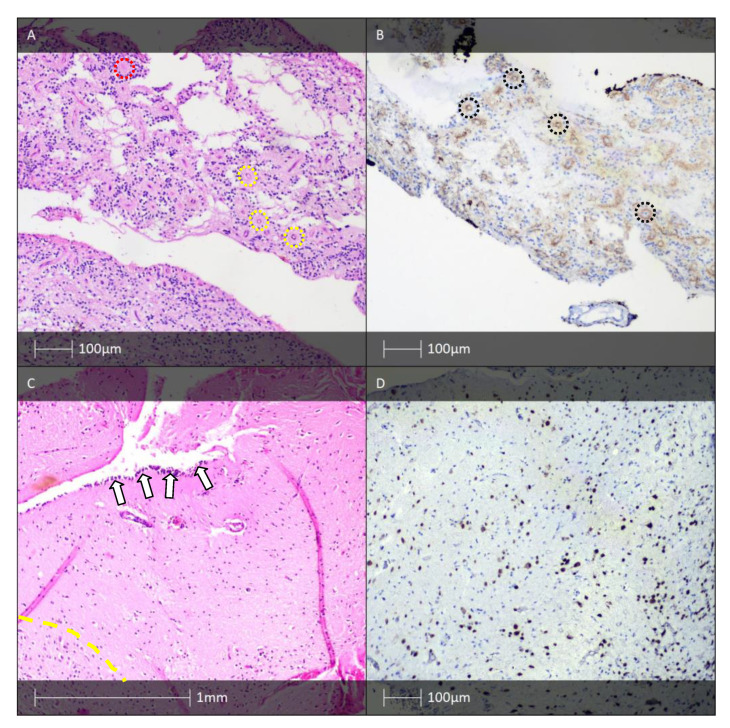
RFGNT and HH. (**A**) RFGNT with perivascular pseudorosettes (yellow dotted circles) and neurocytic rosettes (red dotted circle) (HE). (**B**) RFGNT with perivascular pseudorossettes (black dotted circles, synaptophysin IHC). (**C**) HH—ependymal cells lining the third ventricle (white arrows) and neuronal clustering (yellow dotted line). (**D**) HH—NeuN immunohistochemistry labels the neurones (NeuN IHC).

**Figure 7 diagnostics-13-01311-f007:**
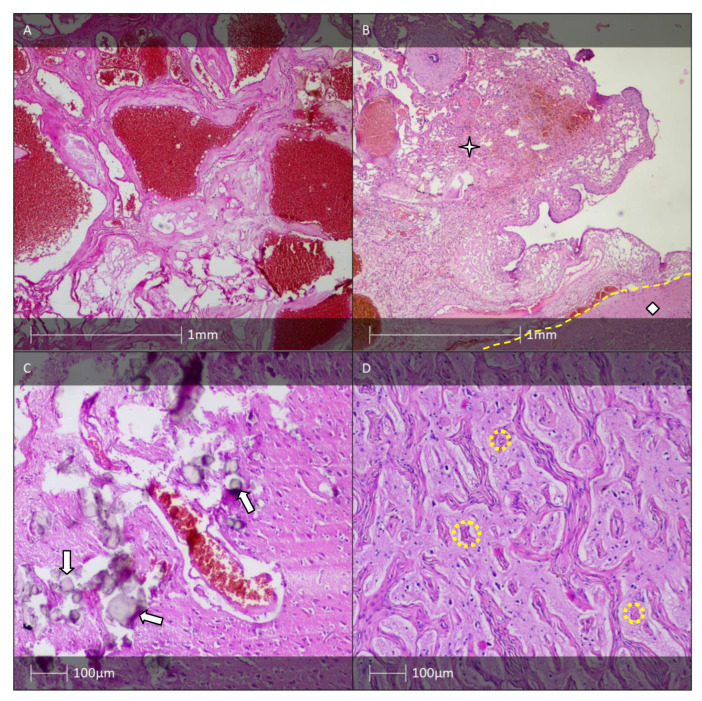
Vascular malformations. (**A**) Cavernous hemangioma (HE). (**B**) Leptomeningeal angiomatosis (asterisk) and with underlying neocortex (white rhombus) (HE). (**C**) Leptomeningeal angiomatosis and microcalcifications (white arrows) in the neocortex of a patient with Sturge–Weber syndrome (HE). (**D**) Meningioangiomatosis—spindle cell meningothelial proliferation around microcirculatory blood vessels (yellow dotted circles) (HE).

**Figure 8 diagnostics-13-01311-f008:**
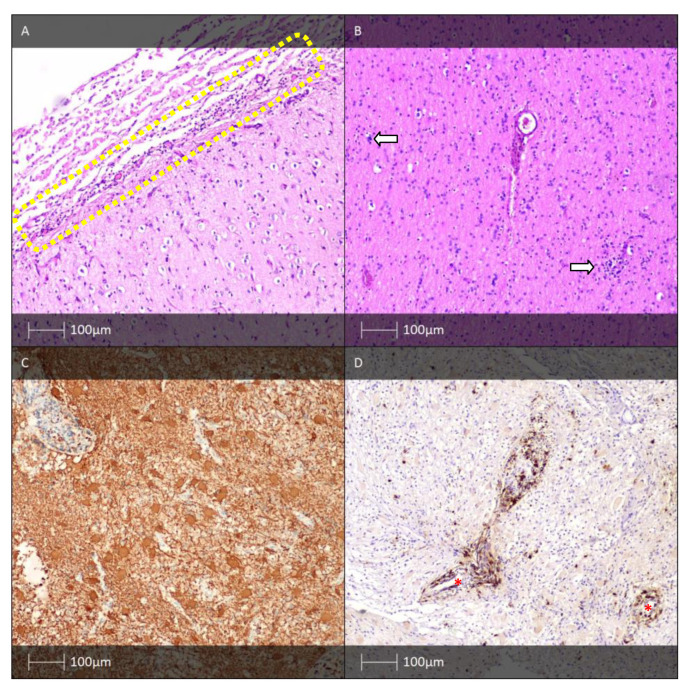
Rasmussen’s encephalitis: (**A**) lymphoid infiltrates in superficial cortical areas (yellow dotted rectangle) (HE); (**B**) microglial cells clustering (white arrows) (HE); (**C**) “cellular” gliosis (GFAP IHC); (**D**) cytotoxic T-lymphocytes surrounding blood vessels (red asterisks—vessel lumen) (IHC-CD8).

**Figure 9 diagnostics-13-01311-f009:**
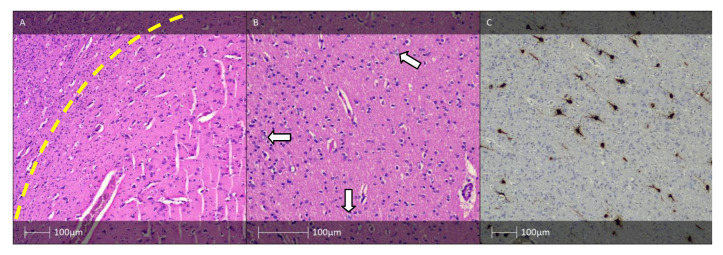
MOGHE. (**A**) The dotted line demarcates the blurred grey–white matter boundary (HE); (**B**) heterotopic neurones in the cerebral white matter (white arrows) (HE); (**C**) immunohistochemical labelling of heterotopic neurones with NeuN (IHC).

## Data Availability

No new data were created or analysed in this study. Data sharing is not applicable to this article.

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
