# Peer review of "Pathomorphological Diagnostic Criteria for Focal Cortical Dysplasias and Other Common Epileptogenic Lesions—Review of the Literature"

_diagnostics, 2023, doi:10.3390/diagnostics13071311_

Round 1
Reviewer 1 Report
This literature review on the diagnostic of focal cortical dysplasia gives clear description of the pathological features and classifications. In my point of view, it is of a significant contribution.
I have only one remark regarding the reference 127, not figuring on the text.
Author Response
We would like to thank Reviewer 1 for taking the necessary time and effort to review the manuscript!
There was a technicall error with the numbering of the reference list which has now been corrected.
Reviewer 2 Report
Comments and suggestion Suggestions for the author:
Define pharmacoresistant epilepsy in simple terms so that readers who may not be familiar with the term can understand the abstract better.
Summarize some of the key findings from the literature on the morphological changes observed in patients with pharmacoresistant epilepsy in a concise manner.
Explain why the new histologically defined pathological entities discussed in the 2022 revision of the ILAE FCD classification are significant and how they differ from previous classifications.
Provide examples of novel treatment strategies based on genetic variants in FCD and explain how they might be implemented in clinical practice.
Include a brief summary of the key findings discussed in the paper in the conclusion to reinforce the main points for the reader.
Suggest specific research directions that could help advance our understanding of FCD in conclusion, such as the potential for genetic studies or imaging studies to shed light on the underlying mechanisms of FCD.
Summarize some of the pathomorphological characteristics of individual forms of FCD briefly in the conclusion to reinforce the main points of the paper.
Provide examples of other epileptogenic lesions associated with FCD and explain how they contribute to the development of pharmacoresistant epilepsy in patients with FCD.
Please cross check the references with manuscript to cite the exact reference.
Thank you.
Author Response
Please see the attachment which includes a point-by-point response to each of the reviewer’s comments.

Reviewer 3 Report
Nice job. In this review, the authors summarized the pathomorphological characteristics of the FCD and other common epileptogenic lesions. The rich illustrations undoubtedly help the reader to understand these pathomorphological diagnostic criteria.
Author Response
We would like to thank Reviewer 3 for taking the time and effort to review our manuscript. We really appreciate the reviewer's positive comments.